

# Temporal and spatial mediated changes in subsurface microbial community assemblages and functions

Madison C. Davis[1]

[1]Department of Cell Biology, Microbiology and Molecular Biology, University of South Florida, Tampa, FL, 33620, USA

*Correspondence to*: Madison C. Davis (madison7@usf.edu)

**Abstract.** Groundwater ecosystems can host different habitats with unique microbial assemblages and functions. Although groundwater microbes are important to subsurface processes, little is known about the drivers of change in these communities. Illumina sequencing and bioinformatic tools were used to examine whether different groundwater zones could have the same patterns of microbial community change over a two-year period. Five different groundwater zones from Hospital Hole, a

stratified sinkhole in west-central Florida, were used in this study since they have been previously shown to host distinct microbial communities. Seasonal patterns of microbial community assemblages and potential metabolic functions were not identified in the sinkhole communities. Different physicochemical parameters correlated to microbial community change within each zone. Local hydrogeology appears to play an important role in subsurface microbial community change since Hurricane Irma and seasonal turnover events did not appear to cause a large perturbation in the microbial communities. Nutrient

availability and local hydrogeochemistry appear to be important drivers of microbial community change in the subsurface.

## 1 Introduction

Both community interactions and hydrochemistry drive change in microbial communities (e.g. Grubisic et al. 2017, Graham et al. 2016), but physicochemistry plays a greater role in seasonal changes in microbial community composition (Gilbert et al. 2012). These drivers can cause short-term (Landesman and Dighton 2011, Landesman et al. 2019, Gunnigle et

al. 2017, Davis et al. 2020) and long-term (Lipson et al. 2004, Siboni et al. 2016, Ward et al. 2017) temporal changes in microbial community structure and function. Temporal variation in microbial assemblages can allow for a new consortium (Rubelmann 2014) or recurring patterns of communal assemblages (Bosshard et al. 2000, Camacho et al. 2000, Dimitriu et al. 2008) when similar physicochemical patterns occur (Bosshard et al. 2000, Ward et al. 2017).

Temporal dynamics of microbial community assemblages vary within and among systems. Some systems have

community variation depending on the location of sampling (e.g. Siboni et al. 2016), whereas other systems can have relatively stable microbial communities on a short-term (e.g. Koizumi et al. 2004, Rogozin et al. 2010) or long-term basis (Dimitriu et al. 2008). There are many environmental factors that drive seasonal changes (Landesman et al. 2019), and how these factors manifest appear to be based on the type of ecosystem. For example, seasonal changes in temperature are known to drive



microbial community assemblages, but this may manifest as changes in water (Siboni et al. 2016, Ward et al. 2017) or soil
temperature (Strickland et al. 2015, Alster et al. 2016).

Perturbation events can cause dramatic changes within microbial communities (Alcocer et al. 1999, Lee et al. 2017, Rubelmann 2014, Menning et al. 2018, Ager et al. 2010), though little is known about how long their impacts may last. Environmental conditions can cause these perturbations, such as seasonal turnover of water columns caused by temperature changes and seasonal weather patterns (Alcocer et al. 1999, Lee et al. 2017, Rubelmann 2014, Menning et al. 2018). While
these perturbations can alter the microbial communities, the assemblages may return (Ager et al. 2010) or a new consortium may form with the same functions (Rubelmann 2014). Microbial communities that incur perturbation events may become resistant to other environmental stressors (Bowen et al. 2011, Bressan et al. 2009).

This study sought to determine whether patterns of microbial assemblages and potential metabolic functions are similar in different zones within a submerged sinkhole over a two-year period. It was hypothesized that seasonal microbial
community changes occurred all zones. This study investigated five different adjacent zones in a stratified sinkhole that had been previously characterized (Davis and Garey 2018) to provide a holistic view of microbial community changes within the subsurface.

## 2 Materials and Methods

### 2.1 Site description, sampling procedures, and Hurricane Irma

Hospital Hole is a submerged, 40-m deep sinkhole in west-central Florida underneath the Weeki Wachee River, FL about 1.4 km inland (Fig. 1). Previous analyses (Davis and Garey 2018) have indicated four unique hydrochemical zones (Fig. 1) within this sinkhole: the surface zone (Weeki Wachee River), the hypoxic zone, the chemocline, and the anoxic zone. Subsequent observations demonstrated that a conduit at 22 m deep contributed water with a unique chemistry to the hypoxic zone, thus the fifth zone (Fig. 1). Other conduits discharging water with similar hydrochemistry and the presence of flowstone
were noted in the hypoxic zone.

Sampling was conducted by trained scientific cave divers under the auspices of the University of South Florida Scientific Diving Program. Five replicate samples per zone were collected in 1L bottles for biological and three replicate samples were collected in 500 mL glass bottles for hydrochemical analysis (sulfide, sulfate, phosphorus, nitrite, alkalinity, and ammonia). Three separate 50 mL conical vials were used to sample for total organic carbon (TOC) and total nitrogen (TN). A
datasonde (OTT Hydromet, Loveland, CO, USA) accompanied divers on each dive to measure pH, salinity, temperature, dissolved oxygen (DO), and depth at 5-s intervals.

The sampling period encompassed two wet seasons and two dry seasons. Hospital Hole was sampled on the following dates: July 11, 2017; September 28, 2017; November 14, 2017; January 16, 2018; April 3, 2018; July 17, 2018; September 18, 2018; December 4, 2018; and April 16, 2019. Hurricane Irma impacted the study region, and the eye passed east of the study
region on September 11, 2017 as a Category 1 hurricane (Cangialosi et al. 2018). The conduit at 22 m deep was not sampled





prior to Hurricane Irma since it was thought to be inactive until September 28, 2017 when divers felt heavy flow from the conduit.

## 2.2 Hydrochemical analyses of water column samples

Water samples were collected by divers, transported on ice, and analyzed in the lab. Total organic carbon and total
nitrogen were analyzed with a Shimadzu TOC-V Analyzer (Shimadzu Scientific Instruments, Kyoto, Japan). Sulfate (method 10248), sulfide (method 8131), alkalinity (method 10239), nitrite (method 10207), phosphorus (method 8190), and ammonia (method 10205) were analyzed using HACH test kits and a HACH DR 3900 spectrophotometer (HACH, Loveland, CO, USA). Protocols from the manufacturer were used. Deoxygenated deionized water was used for all necessary sample dilutions. Monthly rainfall data for Hernando County for each month sampled was obtained from the Southwest Florida Water
Management District (SWFWMD) website.

Statistical analyses of the replicate hydrochemical data were analyzed using Primer v7/Permanova+ statistical software to correlate with community structure data. Hydrochemical data was transformed (log X+1), normalized (subtracted the mean across all samples and divided by the standard deviation of the variable), and clustered using Euclidian distance similarity before correlating with microbial community data (Primer v7/Permanova+).

## 2.2 Microbial community analyses

Water column samples (1 L) were filtered through sterile 0.22-µm filters. Bacterial DNA was extracted aseptically using the Qiagen PowerSoil kit following the manufacturers protocol. The V4-V5 region of the 16S rRNA gene for three replicate DNA samples and was amplified using the Earth Microbiome 515F and 806R primers (Caporaso et al. 2011) adapted for Illumina MiSeq sequencing by Applied and Biological Materials, Inc. (Richmond, BC, Canada). A mock community was
constructed from a six-strain mix (ATCC MSA 3000), Thermococcus gorgonarius (ATCC 700654D-5), and Methanococcus maripaludis (ATCC 43000D-5). Sequences from the mock community samples were analyzed and processed through mothur separately (Schloss et al. 2009) to calculate the sequence error rate. A reference dataset was constructed from the strain information using sequences from Genbank.

Sequencing data for each zone was run separately through mothur software (Schloss et al. 2009) to assemble paired-
end reads and to remove ambiguous sequences, sequences greater than 310 base pairs in length, and archaeal, eukaryotic, mitochondrial, chloroplast, and unidentified sequences. Mothur was used to eliminate chimeras using the VSEARCH algorithm (Rognes et al. 2016) and to create operational taxonomic units (OTUs, ≥97% similarity) using the OptiClust algorithm (Westcott and Schloss 2017). Any OTU with less than 20 sequences were removed from subsequent analyses to avoid potential artifacts caused by rare OTUs (Brown et al. 2015). Rarefaction curves were produced using mothur (Schloss
et al. 2009). The average abundance of each OTU from each date was used for community structure analyses with Primer





v7/Permanova+ statistical software. Principal coordinate analyses (PCO) were used on data that was clustered using Bray-Curtis similarity. Community similarity was determined using the CLUSTER analyses in Primer v7/Permanova+.

The 200 most abundant bacterial OTUs from each zone were investigated for provisional taxonomic identification and functional analyses. The OTUs that were not identified to the genus level by the Ribosomal Database Project (RDP) version 16 reference file (Cole et al. 2014) using mothur (Schloss et al. 2009) were provisionally identified using the nearest identified sequence match from a GenBank query. Those that were at least 80% similar to taxonomically identified sequences in Genbank or RDP were designated as "unidentified". Despite a 97% cut-off for OTU clustering, some OTUs have the same provisional identification, which were combined under the same identification.

The potential metabolic function of each OTU was assigned by a review of the literature for each identified prokaryote. These potential functions include the metabolism of sulfur, nitrogen, carbon, manganese, and iron cycling. To reduce the complexity of the function data, some of these functions were categorized together: microbes capable of sulfur disproportionation or dissimilatory sulfate reduction are considered sulfur reducers; sulfur oxidizers include all microbes that can oxidize sulfur compounds; nitrogen reducers include microbes that carry out denitrification, reduction of nitrate to nitrite, reduction of nitrite to ammonia, dissimilatory nitrate reduction to ammonia (DNRA), and nitrogen fixation; and nitrogen oxidizers include microbes that can utilize nitrification and anaerobic ammonia oxidation.

The relative abundance of sequences with metabolic functions (sequence abundance) was calculated independently in each zone by date. For each date, the number of sequences with a provisional function was divided by the total number of sequences in each date and converted to a percent. The top 200 OTUs were used since they encompass between 60% and 85% of the total sequence abundance.

**3 Results**

Hydrochemical variation within the zones of Hospital Hole were evident (Table 1). Large ranges of concentrations of sulfide in the anoxic (12,456-54,886 µg/L) and chemocline (11-14,079 µg/L), sulfate in the surface (8-529 mg/L), and nitrite within the hypoxic (0.00-0.16 mg/L) zones occurred during this study period. Rainfall was not abnormally high after Hurricane Irma, and sinkhole hydrochemistry does not reflect a drastic change before and after the hurricane (Table 1).

Mothur analyses and OTU clustering of the Illumina sequences revealed a total of 22,487 OTUs representing 5,217,879 bacterial sequences for all five zones analyzed over the two-year study. The sequencing error rate for this dataset was on average 0.09%. Only the surface zone showed clear seasonal patterns when microbial community structure was analyzed by PCO and coded by season (Fig. 2). Communities within each zone were at 5% - 60% similar over the study period, with the least seasonal similarity in the chemocline (5%) and the greatest seasonal similarities in the surface (60%) and anoxic (40%) zones. All zones showed different patterns of community structure (Fig. 2). The identity and function of the 200 most abundant bacterial OTUs in each zone, representing a total of 1,000 OTUs, were analyzed in detail and represent 60%-85% of the total





sequence abundance in each zone. A condensed, literature-based analysis of the potential microbial community is shown in Table 2.

125       While wet and dry season patterns of hydrochemistry were present in the surface and chemocline zones (Table 3), the microbial community structure (Fig. 2) did not show a clear separation between seasons in the zones. Statistical analyses of the microbial community structure determined that seasonal differences in microbial community assemblages were not significant (Table 3). Microbial community function (Table 2) did not show seasonal patterns. A BEST test determined the environmental parameters from Table 1 that best explained changes in microbial community structure from Fig. 2 for each of the five zones (Table 3). While several physicochemical parameters appear to correlate to microbial community changes within

multiple zones, there is not one hydrochemical parameter that correlates to microbial community differences within all zones (Table 3).

      Rainfall did not correlate to changes in microbial community structure (Table 3). Large perturbation events after Hurricane Irma were not identified in the microbial community structure (Fig. 2). Disturbances may have occurred in the potential metabolic functions after this hurricane, but the percent abundance is comparable to later dates (Table 2).

**4 Discussion**

**4.1 Potential causes of microbial community change**

      Reoccurring physicochemical conditions can cause seasonal patterns of microbial community assemblages (e.g. Bosshard et al. 2000, Strickland et al. 2015, Siboni et al. 2016, Ward et al. 2017). Differences between wet and dry season hydrochemistry occurred in the surface and chemocline zones (Table 3), but clear trends in the microbial community structure

(Fig. 2) and estimated metabolic functions (Table 2) were not apparent in the two-year period of this study. The spring-fed surface zone did not experience large temperature changes (Table 1), which likely prevented seasonal mixing events that have been described in other systems (e.g. Saccà et al. 2008, Rubelmann 2014). Considering perturbation events allow for microbial communities to become resistant to environmental stressors (Bowen et al. 2011, Bressan et al. 2009), the lack of seasonal changes could cause subsurface microbial communities to be more susceptible to anthropogenic stressors.

In subsurface ecosystems, nutrient availability may have an important role in inducing microbial community changes. Sulfide concentrations correlated to microbial community changes in the anoxic zone (Table 3). It is likely that the concentrations of sulfide did not elicit changes in the microbial community but was an indicator of changes in the sulfur-reducing community within this zone (Table 2). Availability and changes in the concentration of organic carbon (Table 1) likely caused changes in the community in this zone. Phosphorus concentrations correlated to microbial community changes

in the hypoxic and conduit zones (Table 3). While the input of these nutrients is likely from agricultural runoff or leaky septic systems, these communities provide an analogy to what could occur near injection wells for aquifer storage and recovery. Correlations of hydrochemistry to microbial community change varied between zones (Table 3), suggesting that different groundwater regions may not have the same drivers of change.



## 1.2 Impacts of Hurricane Irma

Tropical storms and hurricanes may affect subsurface microbial communities (Lakey and Krothe 1996, Pronk et al. 2006, Menning et al. 2018) by concentrating large amounts of rainfall to a small region, which can alter flow and discharge within the surrounding aquifer (Menning et al. 2018). In some cases, weather events may not elicit changes in subsurface communities (e.g. León-Galván et al. 2009) since not all storms produce large amounts of rainfall. Hurricane Irma did not produce a large change in microbial community assemblages in the zones of Hospital Hole (Fig. 2). The abundance of sulfur

oxidizers in the hypoxic zone decreased after Hurricane Irma (29%) and returned to their previous abundance after the storm (50%), but similar abundances were observed in July (28%), September (28%), and December (29%) the following year. It is likely that Hurricane Irma did not have as much impact as other tropical storms and hurricanes in the area (e.g. Menning et al. 2018) since Hurricane Irma did not result in abnormal amounts of rainfall (Table 1).

## 1.3 Interactions between distinct hydrochemical zones

There is a general assumption in groundwater management that aquifer microbial communities are homogenous and that communities in different hydrogeochemical regions of aquifers do not interact. The year-round stratification of Hospital Hole allows it to be a model of hydrochemical stratification within the aquifer (Davis and Garey 2018). Similar patterns of microbial community assemblages (Fig. 2B-D) and similar trends in changes of abundance (Table 2) in the zones of Hospital Hole suggest interactions between hydrochemically distinct water bodies in the subsurface. It is important for water managers

to consider how changes in the microbial community of one region may impact the biogeochemistry of other regions.

Interactions between the different zones in Hospital Hole may have occurred (Fig. 3). Phosphorus concentrations correlated to microbial community change in the hypoxic and conduit zones (Table 3). The hypoxic zone is partially sourced from water discharging from the conduit (Fig. 1), so changes in flow, water source, and hydrochemistry of the conduit may affect the microbial communities in the hypoxic zone. Species interactions are generally considered on a scale of a few

micrometers (Armitridge and Jones 2019), but metabolic byproducts of these communities could impact a larger region. For example, the sulfur reducing community within the anoxic zone (Table 2) created high concentrations of hydrogen sulfide (Table 1), which the sulfur oxidizers in the chemocline could utilize (Table 2). Discharge from Hospital Hole into the Weeki Wachee River (Sinclair 1978) may affect the biogeochemistry of the Weeki Wachee River downstream of the sinkhole but was not analyzed in this study. The relationship of changes in microbial communities among these zones suggests that regional

scale features are important to consider for management strategies within karst, which has been noted in other studies (e.g. Scharping et al. 2018).

## 4 Conclusions

The subsurface contains distinct hydrochemical zones with unique microbial community assemblages and functions, and the microbial communities within each of these zones may respond differently to environmental changes. Conduits and

the interaction of different water bodies may add to the complexity of understanding drivers of microbial community change in subsurface ecosystems (Fig. 3). Perturbation events, such as hurricanes, tropical storms, and seasonal turnover, may cause changes in microbial community structure and function if the hydrogeochemistry is altered (Fig. 3). Water managers should be cautious of the introduction of nutrients to the subsurface and the alteration of groundwater flow. Future studies should also consider the ecological impacts these microbial communities may have on large ecosystem processes.

**Data availability**

The sequencing data will be deposited and available in GenBank.

**Author contribution**

MCD and JRG designed the experiments and MCD carried them out and analyzed the bioinformatics. MCD prepared the initial manuscript and JRG helped in revisions. All authors contributed to the manuscript, read, and approved the submitted
version.

**Acknowledgements**

I gratefully acknowledge and thank Tony Green and USF IT support for their assistance in the bioinformatic analysis; James Garey, Robert Scharping, Meredith Snyder, Chelsea Dinon, Victoria Frazier, and the undergraduate volunteers for their assistance in sample collection and hydrochemical analysis; and Jason Gulley, Sarina Ergas, Prahathees Eswara, Bogdan Onac,
Zachary Atlas, Luna Davis, and SWFWMD for their support.

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

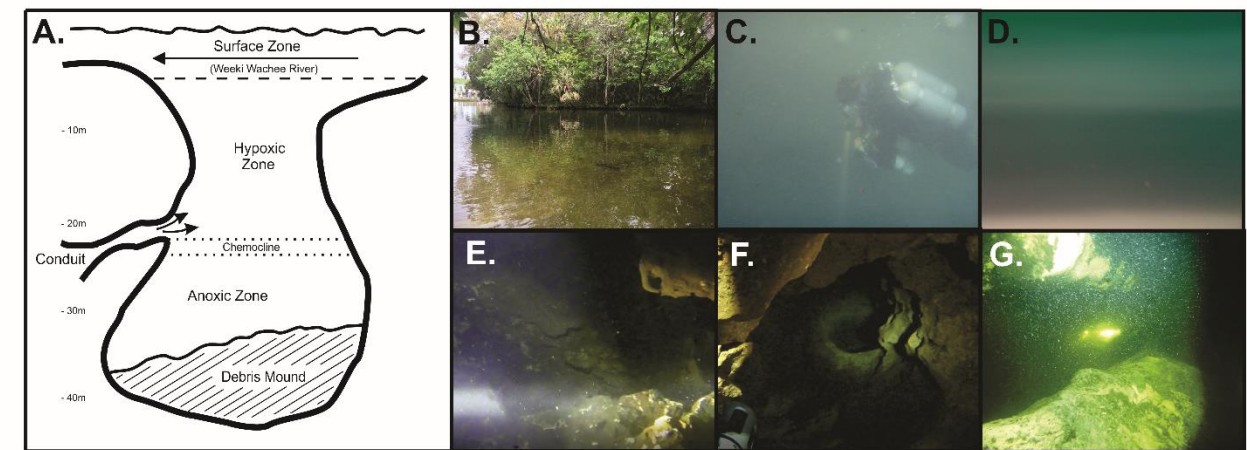

**Figure 1: (A.) Profile view of Hospital Hole site modified from Davis and Garey (2018), as seen Chapter 1. (B.) The surface zone, the**
**Weeki Wachee River, flows over the sinkhole. (C.) Diver in the hypoxic zone of the sinkhole, which extends from 3 m depth to roughly 22 m depth. (D.) The chemocline is a white, cloudy zone that can range from a couple centimeters to a couple meters in**





thickness at roughly 22 m depth, depending on tide. The profile (E.) and internal (F.) view of the conduit is shown. This conduit is at roughly 21 m depth and flows water into the hypoxic and chemocline zones. (G.) The anoxic zone extends from underneath the chemocline at roughly 22 m deep to the bottom of the sinkhole, which is about 40 m depth.

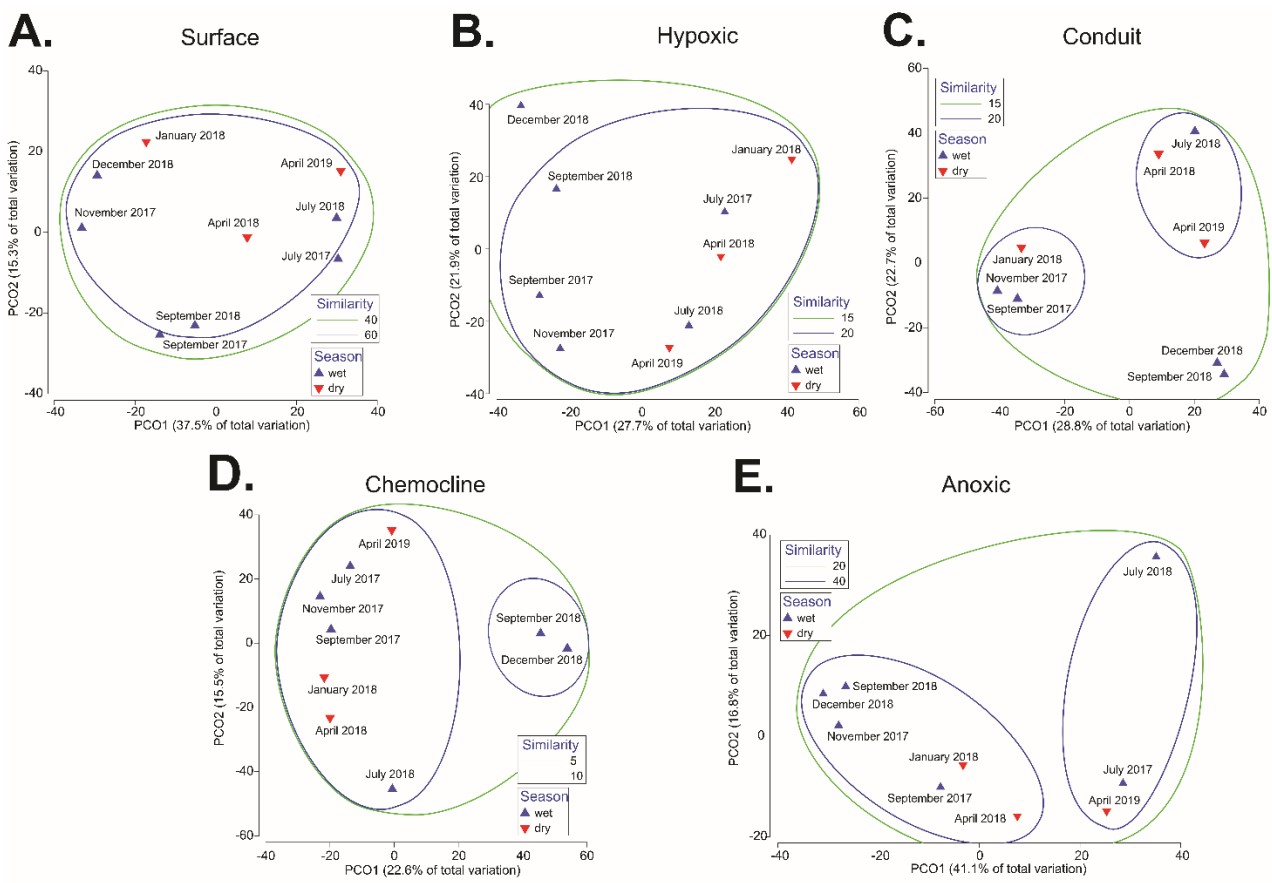


**Figure 2: Principal coordinate analyses of the (A.) surface, (B.) hypoxic, (C.) conduit, (D.) chemocline, and (E.) anoxic zones are shown. The average of three replicate microbial community samples are shown per date and are color coded by season. Similarities of the communities within each zone are also shown.**





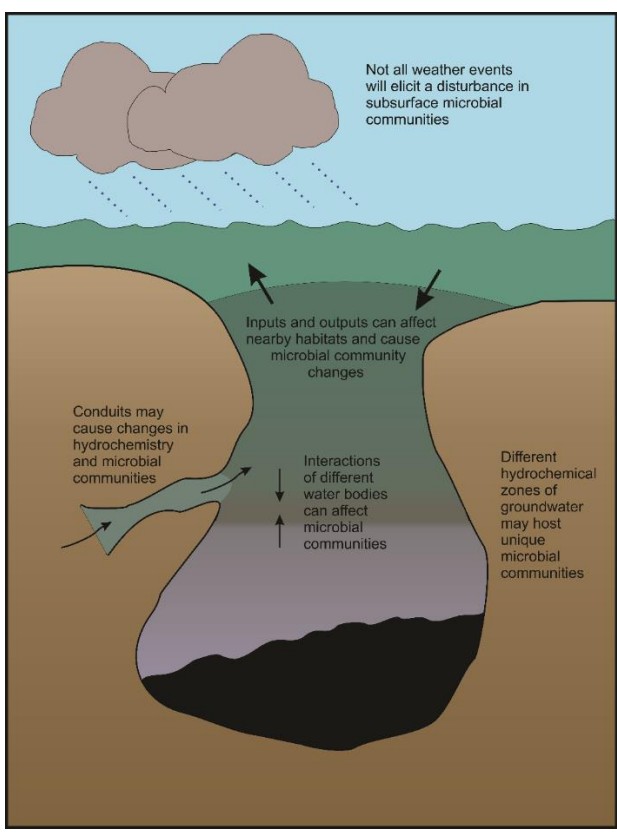

**Figure 3: Some potential drivers of subsurface microbial community changes are shown in relation to how these drivers could impact Hospital Hole.**





| Region | Phyisochemical parameters | Units | Jul-17 | Sep-17 | Nov-17 | Jan-18 | Apr-18 | Jul-18 | Sep-18 | Dec-18 | Apr-19 |
|---|---|---|---|---|---|---|---|---|---|---|---|
| Surface | Phosphorus | mg/L | 0.1 | 0.1 | 0.1 | 0.1 | 0.1 | 0.2 | 0.2 | 0.2 | 0.3 |
| | Ammonia | mg/L | 0.02 | 0.00 | 0.00 | 0.00 | 0.00 | 0.00 | 0.01 | 0.01 | 0.00 |
| | Nitrite | mg/L | 0.01 | 0.00 | 0.00 | 0.00 | 0.00 | 0.00 | 0.00 | 0.00 | 0.00 |
| | Sulfide | µg/L | 2 | 1 | 0 | 1 | 1 | 0 | 0 | 0 | 2 |
| | Sulfate | mg/L | 14 | 8 | 10 | 16 | 18 | 13 | 529 | 19 | 39 |
| | Alkalinity | mg/L | 120 | 117 | 122 | 182 | 150 | 103 | 123 | 108 | 150 |
| | Dissolved oxygen | mg/L | 6.0 | 5.2 | 6.8 | 7.7 | 5.8 | 5.4 | 5.4 | 6.3 | 6.8 |
| | Salinity | ppt | 0.4 | 0.3 | 0.4 | 1.2 | 0.9 | 0.5 | 0.5 | 2.5 | 0.5 |
| | pH | | 7.7 | 7.6 | 7.8 | 7.8 | 7.6 | 7.6 | 7.5 | 7.5 | 8.0 |
| | Temperature | °C | 24.33 | 23.98 | 23.24 | 22.21 | 23.19 | 24.36 | 25.05 | 23.24 | 23.13 |
| | Total organic carbon | mg/L | N/A | N/A | N/A | N/A | N/A | 2.3 | 0.2 | 0.9 | 1.0 |
| | Total nitrogen | mg/L | N/A | N/A | N/A | N/A | N/A | 4.1 | 0.3 | 1.2 | 0.6 |
| Hypoxic | Phosphorus | mg/L | 0.3 | 0.1 | 0.1 | 0.1 | 0.2 | 0.1 | 0.2 | 0.2 | 0.2 |
| | Ammonia | mg/L | 0.05 | 0.06 | 0.00 | 0.00 | 0.01 | 0.00 | 0.06 | 0.02 | 0.00 |
| | Nitrite | mg/L | 0.01 | 0.08 | 0.00 | 0.05 | 0.04 | 0.16 | 0.00 | 0.01 | 0.00 |
| | Sulfide | µg/L | 0 | 1 | 1 | 7 | 1 | 3 | 0 | 0 | 1 |
| | Sulfate | mg/L | 724 | 451 | 529 | 357 | 1 881 | 607 | 425 | 403 | 482 |
| | Alkalinity | mg/L | 114 | 125 | 125 | 139 | 137 | 136 | 137 | 114 | 118 |
| | DO | mg/L | 0.0 | 0.0 | 0.5 | 0.0 | 0.1 | 0.0 | 0.0 | 0.4 | 0.3 |
| | Salinity | ppt | 9.3 | 8.0 | 11.6 | 10.8 | 12.4 | 12.1 | 8.3 | 11.1 | 12.3 |
| | pH | | 7.3 | 7.2 | 7.3 | 7.3 | 7.2 | 7.3 | 7.3 | 7.3 | 7.6 |
| | Temperature | °C | 24.16 | 24.17 | 24.16 | 24.16 | 24.16 | 24.18 | 24.18 | 24.18 | 24.19 |
| | TOC | mg/L | N/A | N/A | N/A | N/A | N/A | 2.1 | 0.5 | 0.8 | 0.7 |
| | TN | mg/L | N/A | N/A | N/A | N/A | N/A | 3.9 | 1.0 | 2.5 | 0.6 |
| Conduit | Phosphorus | mg/L | N/A | 0.1 | 0.1 | 0.1 | 0.1 | 0.1 | 0.3 | 0.2 | 0.2 |
| | Ammonia | mg/L | N/A | 0.0 | 0.0 | 0.0 | 0.0 | 0.0 | 0.1 | 0.0 | 0.0 |
| | Nitrite | mg/L | N/A | 0.0 | 0.0 | 0.0 | 0.0 | 0.0 | 0.0 | 0.0 | 0.0 |
| | Sulfide | µg/L | N/A | 3 | 1 | 3 | 1 | 4 | 0 | 1 | 0 |
| | Sulfate | mg/L | N/A | 341 | 486 | 401 | 847 | 598 | 370 | 403 | 402 |
| | Alkalinity | mg/L | N/A | N/A | 127 | 141 | 150 | 105 | 136 | 116 | 152 |
| | DO | mg/L | N/A | 0.3 | 0.6 | 0.3 | 0.4 | 0.5 | 0.7 | 0.7 | 0.3 |
| | Salinity | ppt | N/A | 7.9 | 10.7 | 11.8 | 12.6 | 12.1 | 8.0 | 10.4 | 12.3 |
| | pH | | N/A | 7.3 | 7.4 | 7.4 | 7.4 | 7.4 | 7.5 | 7.5 | 7.6 |
| | Temperature | °C | N/A | 24.16 | 24.16 | 24.16 | 24.14 | 24.17 | 24.16 | 24.18 | 24.19 |
| | TOC | mg/L | N/A | N/A | N/A | N/A | N/A | 2.0 | 0.1 | 0.6 | 0.9 |
| | TN | mg/L | N/A | N/A | N/A | N/A | N/A | 4.4 | 0.1 | 1.5 | 0.6 |
| Chemocline | Phosphorus | mg/L | 0.3 | 0.3 | 0.2 | 0.5 | 0.2 | 0.3 | 0.2 | 0.3 | 0.4 |
| | Ammonia | mg/L | 0.12 | 0.41 | 0.07 | 0.34 | 0.11 | 0.03 | 0.06 | 0.05 | 0.09 |
| | Nitrite | mg/L | 0.06 | 0.02 | 0.03 | 0.02 | 0.04 | 0.32 | 0.03 | 0.05 | 0.05 |
| | Sulfide | µg/L | 25 | 3 870 | 1 391 | 14 079 | 293 | 30 | 11 | 15 | 475 |
| | Sulfate | mg/L | 513 | 421 | 571 | 543 | 1 038 | 680 | 380 | 570 | 481 |
| | Alkalinity | mg/L | 133 | 146 | 130 | 190 | 148 | 151 | 127 | 116 | 190 |
| | DO | mg/L | 0.0 | 0.0 | 0.2 | 0.0 | 0.2 | 0.0 | 0.0 | 0.3 | 0.1 |
| | Salinity | ppt | 12.4 | 8.8 | 13.1 | 14.2 | 13.9 | 14.0 | 8.5 | 13.3 | 15.8 |
| | pH | | 7.2 | 7.2 | 7.1 | 7.1 | 7.0 | 7.2 | 7.4 | 7.2 | 7.3 |
| | Temperature | °C | 24.49 | 24.18 | 24.22 | 24.16 | 24.16 | 24.19 | 24.18 | 24.18 | 24.16 |
| | TOC | mg/L | N/A | N/A | N/A | N/A | N/A | 2.3 | 0.4 | 1.0 | 2.3 |
| | TN | mg/L | N/A | N/A | N/A | N/A | N/A | 3.3 | 0.9 | 1.9 | 1.1 |





| Region | Phyisochemical parameters | Units | Jul-17 | Sep-17 | Nov-17 | Jan-18 | Apr-18 | Jul-18 | Sep-18 | Dec-18 | Apr-19 |
|---|---|---|---|---|---|---|---|---|---|---|---|
| **Anoxic** | *Phosphorus* | *mg/L* | 0.7 | 0.7 | 0.6 | 0.6 | 0.5 | 0.7 | 1.1 | 0.8 | 0.4 |
| | *Ammonia* | *mg/L* | 0.06 | 0.06 | 0.06 | 0.13 | 0.03 | 0.15 | 0.10 | 0.18 | 0.07 |
| | *Nitrite* | *mg/L* | 0.01 | 0.00 | 0.00 | 0.00 | 0.00 | 0.00 | 0.00 | 0.00 | 0.00 |
| | *Sulfide* | *µg/L* | 37 347 | 43 209 | 37 685 | 34 181 | 23 500 | 12 456 | 54 886 | 44 360 | 31 490 |
| | *Sulfate* | *mg/L* | 1 311 | 1 054 | 1 123 | 800 | 1 675 | 1 067 | 839 | 1 094 | 861 |
| | *Alkalinity* | *mg/L* | 248 | 276 | 247 | 229 | 204 | 284 | 323 | 256 | 222 |
| | *DO* | *mg/L* | 0.0 | 0.0 | 0.0 | 0.0 | 0.2 | 0.2 | 0.2 | 0.1 | 0.0 |
| | *Salinity* | *ppt* | 19.6 | 18.2 | 18.8 | 20.1 | 20.6 | 18.6 | 17.1 | 17.3 | 18.1 |
| | *pH* | | 7.0 | 6.8 | 6.9 | 7.0 | 7.0 | 7.0 | 7.0 | 7.1 | 7.4 |
| | *Temperature* | *°C* | 24.17 | 24.16 | 24.17 | 24.18 | 24.19 | 24.20 | 24.19 | 24.20 | 24.21 |
| | *TOC* | *mg/L* | N/A | N/A | N/A | N/A | N/A | 6.8 | 7.5 | 2.9 | 2.4 |
| | *TN* | *mg/L* | N/A | N/A | N/A | N/A | N/A | 7.6 | 7.9 | 4.4 | 1.5 |
| **All** | *Rainfall* | *cm* | 17.6 | 25.6 | 6.0 | 11.2 | 13.1 | 27.3 | 11.2 | 24.4 | 8.3 |

**Table 2: Hydrochemistry of Hospital Hole.**

| Layer | Potential Metabolic Function | Jul-17 | Sep-17 | Nov-17 | Jan-18 | Apr-18 | Jul-18 | Sep-18 | Dec-18 | Apr-19 |
|---|---|---|---|---|---|---|---|---|---|---|
| **Surface** | Sulfur oxidizer | 3% | 4% | 3% | 27% | 5% | 2% | 2% | 3% | 3% |
| | Nitrogen reducer | 20% | 15% | 12% | 35% | 16% | 19% | 17% | 17% | 17% |
| | Photosynthesis | 1% | 1% | 1% | 1% | 1% | 0% | 0% | 1% | 1% |
| | Iron reducer | 1% | 1% | 1% | 1% | 1% | 1% | 2% | 1% | 2% |
| **Hypoxic** | Sulfur oxidizer | 50% | 29% | 50% | 68% | 68% | 28% | 28% | 29% | 35% |
| | Sulfur reducer | 7% | 3% | 1% | 0% | 0% | 1% | 1% | 1% | 1% |
| | Nitrogen oxidizer | 0% | 0% | 0% | 0% | 0% | 0% | 0% | 8% | 0% |
| | Nitrogen reducer | 73% | 89% | 91% | 69% | 87% | 80% | 93% | 64% | 79% |
| **Conduit** | Sulfur oxidizer | N/A | 29% | 29% | 30% | 45% | 5% | 10% | 17% | 13% |
| | Sulfur reducer | N/A | 1% | 1% | 0% | 0% | 0% | 0% | 0% | 0% |
| | Nitrogen oxidizer | N/A | 6% | 6% | 2% | 3% | 3% | 29% | 24% | 5% |
| | Nitrogen reducer | N/A | 42% | 42% | 32% | 57% | 13% | 16% | 23% | 21% |
| **Chemocline** | Sulfur oxidizer | 92% | 66% | 93% | 80% | 88% | 44% | 54% | 27% | 83% |
| | Sulfur reducer | 7% | 28% | 0% | 0% | 0% | 0% | 0% | 0% | 0% |
| | Nitrogen reducer | 99% | 94% | 99% | 80% | 89% | 83% | 94% | 77% | 89% |
| **Anoxic** | Sulfur oxidizer | 7% | 25% | 37% | 52% | 56% | 11% | 34% | 48% | 8% |
| | Sulfur reducer | 25% | 29% | 24% | 14% | 13% | 7% | 12% | 18% | 45% |
| | Nitrogen oxidizer | 0% | 1% | 0% | 0% | 0% | 0% | 12% | 0% | 0% |
| | Nitrogen reducer | 12% | 40% | 41% | 54% | 65% | 13% | 38% | 51% | 19% |

**Table 2: Potential metabolic functions in five zones of Hospital Hole.**






| Statistical analyses | | Surface | Hypoxic | Conduit | Chemocline | Anoxic |
|---|---|---|---|---|---|---|
| **PERMANOVA p-values** | *Seasonal hydrochemistry* | 0.044 | 0.082 | 0.279 | 0.044 | 0.082 |
| | *Seasonal microbial communities* | 0.494 | 0.25 | 0.426 | 0.527 | 0.275 |
| **BEST analysis** | $p_s$ | 0.343 | 0.417 | 0.622 | 0.412 | 0.46 |
| | ***Combined environmental variables*** | nitrite, total organic carbon | phosphorus, sulfide | phosphorus | nitrite, sulfide, dissolved oxygen, total organic carbon, water elevation | sulfide |

**Table 3: Hydrochemical correlates to microbial community structure within each zone.**