# Peer review of "Temporal and spatial mediated changes in subsurface microbial community assemblages and functions"

_Biogeosciences, 2020_

## Referee Comment (RC1) · Anonymous Referee #1 · 30 Jul 2020

This study "Temporal and spatial mediated changes in subsurface microbial community assemblages and functions" combined geochemical measurements and molecular analyses to relate microbial community assemblages and functions with environmental parameters. I find this manuscript certainly interesting, but the presentation and interpretation of data needs to be significantly improved. The presented data is thin to some extent. The current results and discussion are more like a report rather than a research article. The same author published one paper of similar study in journal Water in 2018 https://www.mdpi.com/2073-4441/10/8/972?type=check_update&version=1,which was better presented than the current one.

What are the compositions of microbial communities in different zones?

Fig. 2 I would compare microbial community of different zones in one graph rather than in 5 separate graphs

Tab.2 Many microbes can conduct multiple functions, for examples, one can be Sulfur oxidizer, Nitrogen reducer or Iron reducer at the same time, how did you manage to separate them out?

---

## Referee Comment (RC2) · Anonymous Referee #2 · 31 Jul 2020

In this study, the author analyzed microbial communities from a sinkhole from five different hydrochemical zones and nine time points during a two-year-period. Although hydrochemical parameters showed seasonal patterns, the author did not observe such patterns for the microbial communities. In general, this is an interesting question and a comprehensive data set. However, I have some concerns regarding the analysis, presentation and interpretation of the data. The motivation of this work is not fully clear from the introduction. The introduction is very general and does not provide sufficient information about microbial communities or microbial community dynamics in the subsurface, which is the subject of this study. Moreover, it is not always clear for which environment previous observations of community dynamics are reported, and to what

extent they can be compared to or are relevant for the subsurface system investigated in this study. Since weather extremes are also of interest in this context, they should also be introduced in the introduction. I also wonder if the number of sampling time points is high enough to identify seasonal patterns. It is also not clear what kind of seasonal pattern the author expected to find. In what way were microbial communities expected to change, and what could be key drivers in this sink hole system? This should be worked out in more detail in the introduction. Moreover, the author states that seasonal patterns existed for hydrochemical parameters, however, these data are only provided as tables. Here, a graphical display confirming the seasonal patterns would be helpful. Despite the large set of microbial data that seems to be available, the results section is rather short, and there might be more potential in the data than the author makes use of. It would also be interesting to learn more about which microbial groups are dominant in these sinkhole communities. Finally, the author should make more clear what is the novelty of this study. Some of the (hydrochemical) patterns presented appear to be similar to stratified lakes. What is special about the sink hole system? The conclusions are very general and broad and are not directly derived from the results of this work.

Specific comments: The title is unclear: should this rather be "temporal and spatial changes. . .."? The information in the last two sentences of the abstract is slightly redundant. Please consider rephrasing.

l. 76: what filter material was used? Please add the information. l. 82-83: This is unclear. Was this reference data set used for the community data from the sinkhole or for the mock community? l. 84: Why were sequence data from each zone analyzed separately with the Mothur pipeline? I would suggest analyzing all sequence data together and then only run statistical analyses with selected data for within-zone comparisons. Comparison of data across the different zones could add additional value to the results. l. 96-98: This is unclear. Does the author mean that sequences with less than 80% similarity were termed "unidentified"? l. 99-105: This approach seems a bit complicated given the fact that programs such as PICRUSt are available that would predict functions based on 16S rRNA gene sequences using comprehensive databases. Why did the author not try such an approach? l. 112: Why are sulfide concentrations not expressed as mg/L? l. 129-131: Please add information here which hydrochemical parameters appeared to be important in the different zones. l. 133-134: This sentence is unclear. How is percent abundance linked to potential metabolic functions? l. 151: It is not clear how this analogy would work. The first Table 2 should be table 1. Table 2: The terms "nitrogen oxidizers" and "nitrogen reducers" are not common. Please use more specific categories such as nitrate reducers, nitrifiers etc.

---

## Author Comment (AC1) · 31 Jul 2020

I'd like to thank Reviewer 1 for their review and comments.

The Davis and Garey 2018 paper sought to characterize the microbial stratification in distinct hydrochemical layers to provide a "snapshot" of microbial stratification within the study site, Hospital Hole. This study sought to determine whether the distinct microbial and hydrochemical layers identified in the 2018 study have similar patterns of microbial community change over a two year time period (summarized in lines 6-11). The principal coordinate analyses allowed for the determination of structural changes within the microbial communities during the study period. Potential metabolic functions

were used in addition to the principal coordinate analyses to determine if there were large shifts in functions (e.g. sulfur oxidizers to sulfur reducers) that may not be present at the taxonomic resolution, and to eliminate the complexity of analyzing changes for thousands of operational taxonomic units for five different groundwater regions. The taxonomic compositions may be added to supplemental materials at the editor's discretion.

The microbial communities were analyzed separately by layer based on the Davis and Garey 2018 initial findings, which is why there are 5 separate principal coordinate analyses plots. Additionally, the changes in structure over time were more easily shown between layers separately than on one plot.

Calculations for potential metabolic functions are described on lines 99-108, which highlights that each function is calculated separately. As mentioned by Reviewer 1, many bacteria can have more than one function (e.g. coupling sulfur and nitrogen cycling). This is why the functions were calculated separately and the percent abundance of functions does not total 100%. This can be clarified in text before publication.

Could Reviewer 1 please clarify how this manuscript may be presented more clearly? Specifically, which sentences in the text should be altered? Of these sentences, which lines and how could they be altered to avoid sounding like a report? Also, would the reviewer please clarify how these sentences could be improved so they do not sound "thin"?

I appreciate the feedback from Reviewer 1, and any clarifications provided to help improve this iteration of the manuscript before publication.
* * *

---

## Referee Comment (RC3) · Anonymous Referee #1 · 1 Aug 2020

The cooments from review are not well addressed, and the author's rely didn't show strong willingness to improve the manuscript. So my suggestion is diect: reject.

---

## Author Comment (AC2) · 1 Aug 2020

I'd like to thank Reviewer 2 for their review and comments.

Subsurface microbial community dynamics are poorly characterized, especially in the coastal regions of Florida. The microbial community dynamics of other systems, including meromictic lakes, were used as a comparison of what could be occurring in the subsurface. The seasons referred to are "wet" and "dry", and were chosen based on the comparison to the literature of other systems. It was expected that rainfall, which would cause changes in hydrology, would impact the microbial communities and would therefore show a distinct wet and dry season microbial community in each layer, as was

seen in Davis et al. 2020 (PLOS ONE, doi: 10.1371/journal.pone.0232742). It was unknown whether the changes in hydrology would cause similar or different patterns in the microbial community assemblages. Hospital Hole is unique in that it is a spring, and conduits from the aquifer discharge into the sinkhole. Davis and Garey 2018 (Water, doi: 10.3390/w10080972) suggested this sinkhole could be analogous to the surrounding aquifer. These items can be clarified in the introduction before publication, which may help elucidate the novelty of this study.

The microbial groups that were dominant were described in Davis and Garey 2018, which provided a comparison of the different microbial communities in each layer, and this study focused on how these groups may change in relation to each other.

Would Reviewer 2 please clarify what is meant by a graphical display of seasonal hydrochemistry, and other ways to analyze the data?

The specific comments with line numbers are addressed below, and can also be clarified before publication: 76 - the filter brand can be added to identify the filter 82-83 - a reference dataset was made for the mock community to calculate the error rate 84 - Davis and Garey 2018 established the differences in the microbial communities in this sinkhole. Due to the large dataset, it was difficult to analyze all of these samples together despite using a cluster network. It was decided, based on the 2018 study, to separate these out by layer to see how the patterns of change compared to each other. 96-98 - if the GenBank queries were less than 80% similar, the taxa was considered "unidentified" 99-105 - PICRUST and other predictive functioning profiles were considered, but these tools may not accurately characterize the extremophiles present within the subsurface. This discussion can be added to the manuscript before publication. 112 - ug/L was used because low concentrations (e.g. 11 ug/L) were present, but can be converted to mg/L before publication 129-131 and 133-134 - these sentences will be clarified before publication 151 - The introductions of nutrients, rather than the hydrology, may be driving microbial community changes in the Weeki Wachee River system. Local management agencies have identified that this springshed has elevated

nutrients. The introduction of nutrients to the subsurface, including those from aquifer storage and recovery programs, may cause changes in the microbial communities in the aquifer. This sentence can be expanded to highlight this discussion.

I thank the reviewer for noting the typo in Table 1. The "nitrogen oxidizers" and "nitrogen reducers" include multiple potential metabolic functions, as noted on lines 99-105.

I'd like to thank Reviewer 2 again for their comments, and appreciate their recommendations which can be included before the publication of this manuscript.

---

## Author Comment (AC3) · 1 Aug 2020

My apologies if I did not appear willing to make the suggested changes. I was asking for clarification on some of the points to ensure the limitations were addressed as the reviewer intended. I will incorporate the comments from both reviewers into the revised version.